# Rapid declines in systolic blood pressure are associated with an increase in pulse transit time

**Sebastian Grøvdal Schaanning[1], Nils Kristian Skjaervold[1,2]***

**1** Department of Circulation and Medical Imaging, Norwegian University of Science and Technology, Trondheim, Norway, **2** Department of Anaesthesia and Intensive Care Medicine, Trondheim University Hospital, Trondheim, Norway

* nils.k.skjervold@ntnu.no

## Abstract

### Background

The correlation between pulse transit time and blood pressure has been proposed as a route to measure continuous non-invasive blood pressure. We investigated whether pulse transit time trends could model blood pressure trends during episodes of rapid declines in blood pressure.

### Methods

From the Medical Information Mart for Intensive Care waveform database we identified substantial blood pressure reductions. Pulse transit time was calculated from the R-peak of the electrocardiogram to the peak of the arterial pulse waveform. The time-series were processed with a moving average filter before comparison. Averaged, continuous heart rate was also analysed as a control. The intra-individual association between variables was assessed per subject using linear regression.

### Results

In the 511 patients included we found a median correlation coefficient between blood pressure and pulse transit time of -0.93 (IQR -0.98 to -0.76) with regression slopes of -1.23 mmHg/ms (IQR -1.73 to -0.81). The median correlation coefficient between blood pressure and heart rate was 0.46 (IQR -0.16 to 0.83). In supplementary analysis, results did not differ substantially when widening inclusion criteria, but the results were not always consistent within subjects across episodes of hypotension.

### Conclusions

In a large cohort of critically ill patients experiencing episodes of rapid declines in systolic blood pressure, there was a moderate-strong intra-individual correlation between averaged systolic blood pressure and averaged pulse transit time. Our findings encourage further investigation into using the pulse transit time for non-invasive real-time detection of hypotension.

**Data Availability Statement:** The waveform data analysed during the current study is available in the MIMIC-III Waveform Database Matched Subset, found at https://archive.physionet.org/physiobank/database/mimic3wdb/matched/. The associated clinical data is available in the MIMIC-III Clinical

Database, found at https://physionet.org/content/mimiciii/1.4/.

**Funding:** SGS was internally funded within Trondheim University Hospital and the Norwegian University of Science and Technology. The funding body did not participate in the design, collection, analysis, interpretation or writing of the study/manuscript. NKS receives postdoctoral funding from Central Norway Health Authority, Grant nr 46056918

**Competing interests:** NKS is chief medical officer and shareholder in Moon Labs, a medical-technology company that is prototyping a wearable biosensor; the company is currently not working with continuous blood pressure monitoring. SGS declares that he has no competing interests. This does not alter our adherence to PLOS ONE policies on sharing data and materials.

# Introduction

Blood Pressure (BP) is an essential vital parameter that is monitored in most if not all hospitalized patients. BP measurements guide inpatient treatment of hypertension, act as a marker of hemodynamic status and severity of illness and can be used for cardiovascular risk stratification. The detection of acute hypotension is of particular interest, as this may signify the onset of circulatory shock, a feared clinical entity that requires prompt hemodynamic support and diagnostic workup [1]. *Continuous* BP-monitoring, however, is currently reserved for patients in whom the invasive insertion of an arterial line can be justified. In practice, this means that for most hospitalized patients, BP is monitored using intermittent cuff-based measurements. The interval between measurements may be several hours, and a recent publication found that a substantial proportion of significant BP-perturbations may go undetected with standard monitoring [2]. This implies that continuous non-invasive BP-monitoring (cNIBP) could benefit patients who are not candidates for arterial line placement.

Various techniques for cNIBP have been proposed, and some integrated systems have been developed. Perhaps most known is the vascular-unloading technique, originally proposed by Marey, and later by Shirer and Penaz [3–5], and employed by Finapres® devices. In this method, a finger cuff is combined with a measure of finger blood volume using photoplethysmography (PPG). By varying the finger cuff pressure to keep the PPG signal constant, a pressure wave can be obtained that approximates the arterial pressure waveform [6]. Another technique for cNIBP is arterial tonometry, which aims to reproduce the arterial waveform based on the transcutaneous displacement produced by a pulsating artery. The method employs an external pressure transducer placed over a superficial artery, typically the radial artery [7]. A third technique which has been subject to much research is utilization of pulse wave velocity (PWV) and the related distance-invariant pulse transit time (PTT). This concept can be rooted in two physio-mathematical relationships, namely the Moens-Korteweg equation [8] and the relationship between elastic modulus and pressure, as empirically demonstrated by Hughes et al. [9], among others. In summary, the combination of these models entails that for an elastic tube the PWV is proportional to the elastic modulus of the tube wall, which in turn is proportional to the pressure within the tube. A further elaboration on these relationships is outside the scope of this article.

PWV and PTT, have been studied extensively for their relationship to BP, dating back to the 1920s. In an oft-cited work from 1976, Gribbin and colleagues demonstrated excellent intra-subject correlation between PWV and transmural arterial pressure, in a setup with external pressure manipulation [10]. Several papers on the relationship between PTT and BP followed. The PTT was dissected into the pre-ejection period (PEP) and the vascular transit time (VTT), with the VTT corresponding to the PWV measured by Gribbin. Researchers generally found an inverse correlation between PTT and BP, but there were conflicting results regarding the contribution of the PEP and the VTT to this relationship [11, 12]. With increased focus on patient monitoring, interest in the subject has thrived again in the last decades. Several researchers have investigated PTT for BP-estimation, typically measuring PTT from the electrocardiogram (ECG) R-peak to the PPG signal at the finger. A review of these efforts was recently published by Ding and Zhang [13]. Researchers have also employed machine learning methods, such as deep neural networks, to find additional PPG-features that may complement or replace PTT for BP-estimation [14].

A popular source of data for PTT-research is the publicly available Medical Information Mart for Intensive Care (MIMIC) database, a collection of waveform data from thousands of patients admitted for intensive care [15, 16]. A summary of studies using MIMIC data to evaluate PTT for BP-estimation was recently presented by Liang et al., along with their own

original research on the database [17]. In short, a negative correlation between PTT and BP is reported by all authors, but patient selection and mode of analysis differs substantially across studies. Liang, like others, noted issues with waveform synchronicity, limiting the suitability of the MIMIC database for analysis of PTT. One major issue is related to the sampling of the PPG-signal, which is usually employed as the endpoint for PTT calculation. Analysing MIMIC data, Zhu et al. noted that the PPG-derived PTT is afflicted by a sawtooth-shaped artefact, with a period of approximately 100 seconds [18]. A similar sawtooth artefact was later demonstrated in the PPG-derived PTT by another group of researchers using a separate database [19]. Bennis et al. observed the same artefact in data from a neonatal ICU, and consequently used simulated data to show that the artefact could be attributed to post-processing of the PPG signal in the Masimo module of the Philips patient monitor [20]. This artefact substantially limits the utility of using previously collected PPG-data to calculate PTT.

While there are many publications on the relationship between PTT and BP, it is our experience that these fall into one of two categories: (1) Small scale experimental studies in healthy volunteers or (2) Studies on database materials with unclear criteria for waveform selection and lack of reporting on the clinical characteristics of the subjects. Most investigations have included a low to moderate number of subjects. In our opinion, the primary potential of cNIBP lies in the early detection of hemodynamic instability, i.e. hypotension, in hospitalized patients without an arterial line. To our knowledge, no one has systematically investigated the relationship between PTT and BP during the onset of hypotension in hospitalized patients. We hypothesize that a moderately rapid ($\leq$ 15 minutes) reduction in SBP from $\geq$120 mmHg to $\leq$ 90 mmHg is accompanied by a predictable change in PTT. Furthermore, where reference data from an arterial line is available, most investigators have tested the ability of PTT to model BP on a beat-to-beat basis. While cNIBP of such resolution may seem enticing, one could argue it may be counterproductive to attempt to capture beat-to-beat fluctuations in BP. Firstly, this places great demands on the precision of the model. Secondly, we believe it is far more clinically relevant to capture short term trends in BP than beat-to-beat fluctuations. For this reason, we chose to perform the primary analysis on time-averaged data to assess short-term BP-trends rather than beat-to-beat fluctuations. Due to issues associated with the PPG-signal in the MIMIC database, we chose to use the Arterial Blood Pressure (ABP) waveform as the endpoint for calculation of PTT.

## Materials and methods

All data preparation and analysis was performed using the Python Programming Language, version 3.7 [21].

### Ethics approval and consent to participate

All data was sourced from the freely accessible MIMIC-III waveform database and the associated clinical database. The following information is provided by representatives for the MIMIC-III project: "The project was approved by the Institutional Review Boards of Beth Israel Deaconess Medical Centre (Boston, MA) and the Massachusetts Institute of Technology (Cambridge, MA). Requirement for individual patient consent was waived because the project did not impact clinical care and all protected health information was deidentified" [15].

### Subject selection

Data was obtained from the MIMIC-III waveform database, matched subset. The matched subset includes 10,282 subjects for whom waveforms have been matched to clinical records. All subjects in the matched subset were eligible for inclusion. For each subject, several

waveform records exist, sometimes exceeding 1000. The records are principally continuous with one another, but were split at time points where signal modalities and/or signal gain were changed. For the purpose of this study, an automated script selected the longest record per subject, for which both ECG and ABP signals were available, for further analysis.

### Event identification and inclusion

An event was defined as an instance where the averaged systolic blood pressure ($SBP_{AV}$) decreased to $\leq 90$ mmHg within 15 minutes of $SBP_{AV}$ exceeding or equaling 120 mmHg. For each record, a scoping algorithm identified candidate segments based on the event definition. Each candidate segment was then evaluated by a series of algorithms to determine if the signal quality was sufficient to determine all peaks needed for parameter extraction. A segment was excluded if one or more of the following features were present:

i.  Max $SBP_{AV} < 120$ mmHg or nadir $SBP_{AV} > 90$ mmHg

ii.  Segment length $< 21$ heartbeats

iii.  ECG signal quality index (ESQI) $< 0.9^{*}$

iv.  Presence of undetectable/ambiguous peaks

   $^{*}$A score from 0–1, primarily based on intercorrelation between QRS-complexes.
   One segment was selected from each subject whose record contained one or more eligible segments. For subjects who had multiple eligible segments, the segment with the highest ESQI was included for primary analysis, and the segment with the second highest ESQI was retained for supplementary analysis.

### Variable extraction from viable segments

Peak detection for ECG signals was performed using the BioSPPy library, version 0.6.1 [22]. ABP peaks were found based on the location of previously detected R-peaks, using the find_-peaks algorithm from the SciPy library, version 1.3.1 [23]. PTT, SBP and HR were then extracted from the data, and moving averages were calculated. The latter was achieved using a 1$^{st}$ order Savitzky-Golay filter with a width of 21 samples. In short, this entails fitting a straight line through the preceding 10 and following 10 heartbeats in order to determine the value at each heartbeat. Extracted variables and their definitions are shown in Table 1. A cutout from one of the segments, providing a visualization of how the PTT-metric is calculated is shown in Fig 1.

**Table 1. Variables extracted from waveforms.**

| Extracted variable | Definition |
| --- | --- |
| PTT-RA | Time interval from the nth ECG R-wave to the nth ABP-peak |
| RR | Time interval between subsequent ECG R-waves |
| HR | (Sample rate*60) / RR |
| SBP | ABP Signal value at detected ABP-peaks |
| PTT-RA$_{AV}$ | Moving average of PTT-RA |
| HR$_{AV}$ | Moving average of HR |
| SBP$_{AV}$ | Moving average of SBP |

The table contains definitions of variables that were used for analysis. See also Fig 1.

## Data analysis

In order to evaluate the endpoint of the study, linear regression was chosen. For each included segment, two separate analyses were performed:

$$SBP_{AV} = \alpha_1 \times PTT - RA_{AV} + \beta_1 \tag{1}$$

$$SBP_{AV} = \alpha_2 \times HR_{AV} + \beta_2 \tag{2}$$

Performing linear regression on averaged data is somewhat unorthodox. In order to verify that the results are not simply a result of the employed methodology, analysis (2) was included as a "negative" control. Reductions in SBP tend to cause reflexive tachycardia, but may also be secondary to a reduction in HR. Thus, the relationship between HR and SBP is not expected to be consistent across subjects. Ultimately, PTT needs to outperform HR in predicting SBP for the results to be considered meaningful.

## Results

### Summary of inclusion and descriptive statistics

Potential events were identified for 2729 subjects. Of these, 511 subjects experienced events without missing/ambiguous peaks. From each of these 511 subjects, one event was included for analysis. Datasets outlining the specific waveform segments that were included for the main analysis can be found in the S1 File. An overview of the inclusion process is shown in Fig 2. Summaries of subject- and event- characteristics are given in Tables 2 and 3.

### Results from regression analysis

Median correlation coefficients (r) obtained from linear regression versus $SBP_{AV}$ were as follows: $PTT-RA_{AV}$ -0.93 (IQR -0.98 to -0.76), $HR_{AV}$ 0.46 (IQR -0.16 to 0.83). For 36 subjects (7.0%), correlation coefficients between $SBP_{AV}$ and $PTT-RA_{AV}$ were positive. A frequency histogram of correlation coefficients is presented in Fig 3.

Regression slopes for the relationship between $SBP_{AV}$ and $PTT-RA_{AV}$ displayed a median of -1.23 mmHg/ms (IQR -1.73 mmHg/ms to -0.81 mmHg/ms). The distribution of slopes is shown in Fig 4.

Fig 5 shows individual data for the subject whose combination of correlation coefficients was closest to the median. Fig 5A–5C shows the time series of extracted parameters, while associated regression plots are shown in Fig 5D–5F. The alignment of data points in vertical bands in Fig 5D illustrates the limited temporal resolution of the PTT metric at a sample rate of 125Hz. Fig 5E and 5F illustrate a side effect of comparing averaged time series: Since subsequent data points are conditioned upon each other, temporal changes can be inferred as a gradual walk over time, even in the absence of a temporal axis.

### Supplementary analysis

In addition to the main analyses, three supplementary analyses were performed to gauge the robustness of the main findings.

1. Linear regression was performed on the raw data for SBP and PTT-RA, without applying any prior averaging.

2. For those subjects included in the primary analysis who experienced more than one event, comparative analysis was performed on the secondary event.

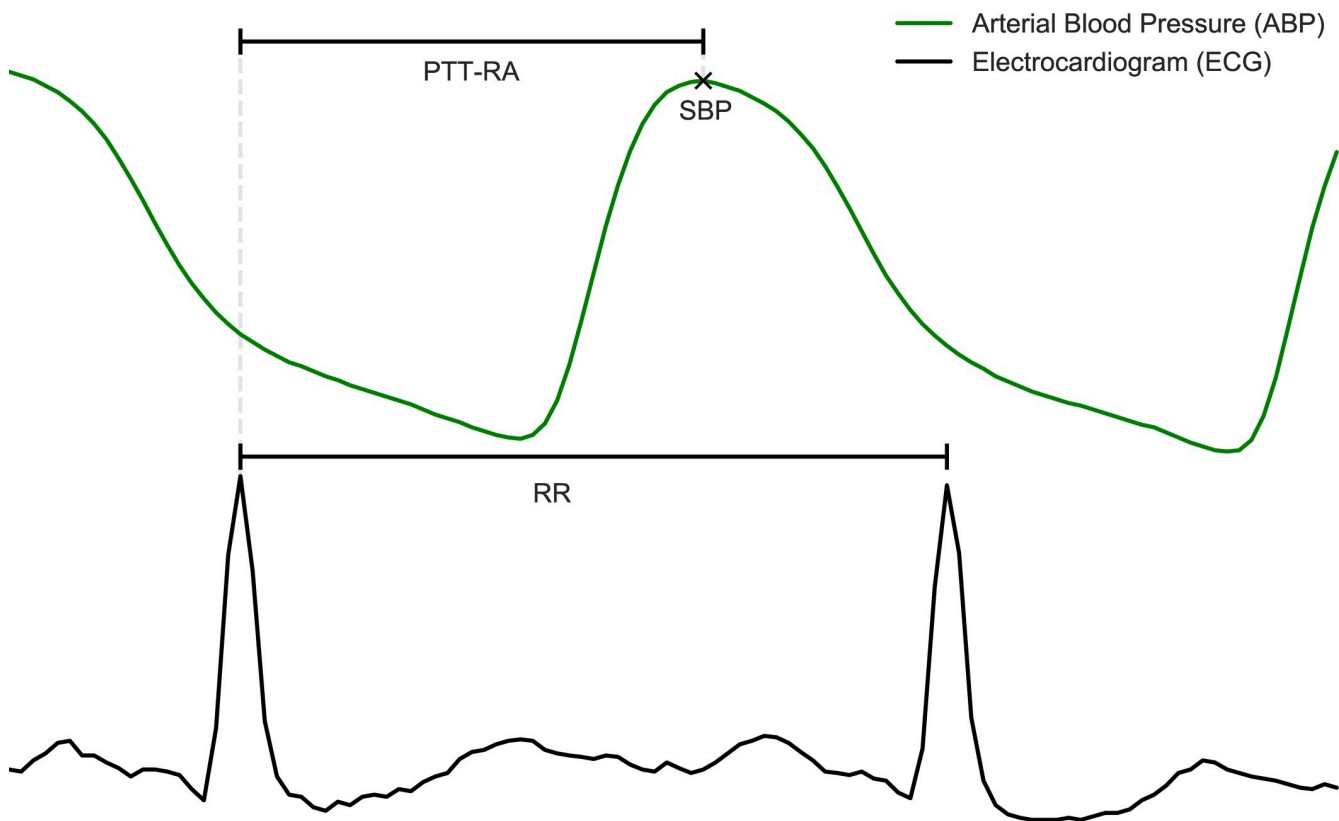

**Fig 1. Illustration of extracted variables.** The figure displays a cut-out of a waveform time series from an arbitrary subject. Annotations define the points and intervals that were extracted from waveforms and used for analysis.

3. The main analysis was repeated with less stringent inclusion criteria, accepting subjects who had < 5 missing/ambiguous peaks (contrary to 0 for the main analysis)

**(1) Linear regression without averaging.** Regression analysis on raw data without averaging gave the following results.

SBP vs. PTT-RA correlation coefficient: Median -0.76 (IQR -0.88 to -0.51)

SBP vs. HR correlation coefficient: Median 0.21 (IQR -0.06 to 0.57)

SBP vs. PTT-RA regression slopes: Median -0.81 mmHg/ms (IQR -1.08 to -0.56)

**(2) Analysis of secondary events.** 215 subjects experienced at least one secondary event that met the inclusion criteria. Regression analysis per subject for secondary events gave the following results:

$SBP_{AV}$ vs. $PTT\text{-}RA_{AV}$ correlation coefficient: Median -0.92 (IQR -0.97 to -0.75)

$SBP_{AV}$ vs. $HR_{AV}$ correlation coefficient: Median 0.53 (IQR -0.16 to 0.87)

$SBP_{AV}$ vs. $PTT\text{-}RA_{AV}$ regression slopes: Median -1.24 mmHg/ms (IQR -1.72 to -0.93)

Regression slopes for primary and secondary segments were also compared in a paired within-subject fashion. Bland-Altman analysis revealed a mean bias of 0.07 mmHg/ms (95%

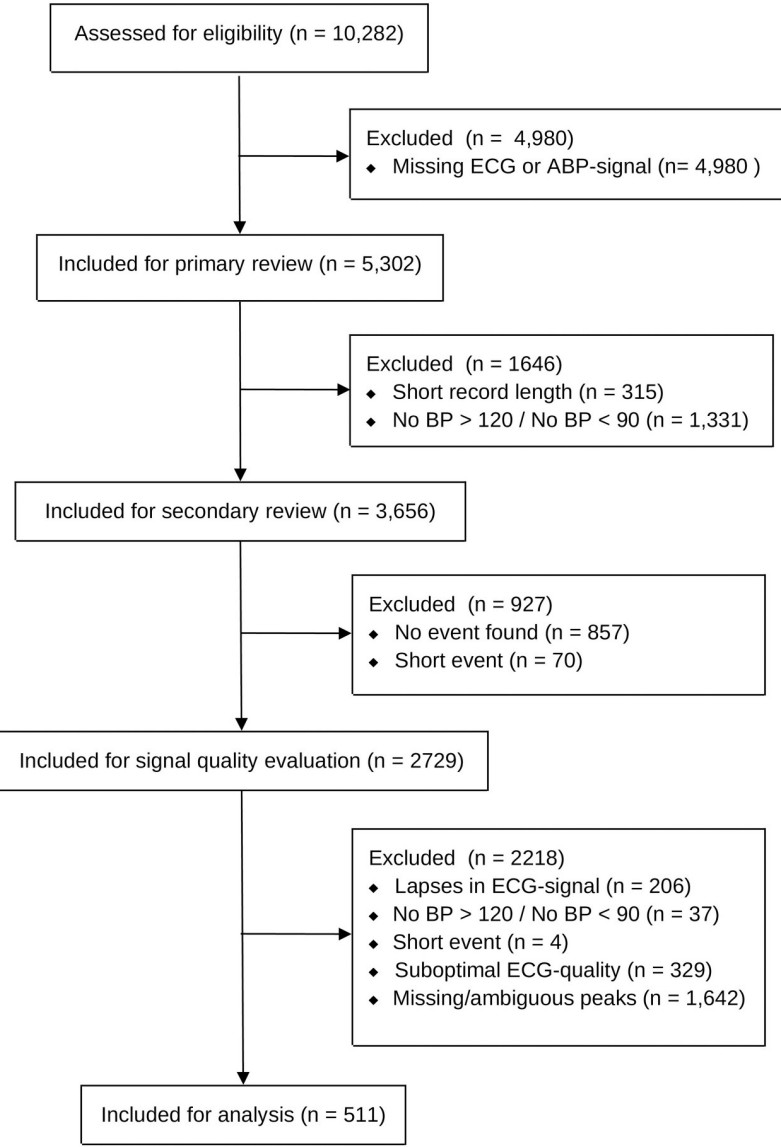

**Fig 2. Overview of included and excluded subjects.** The flow chart displays the process to assess subject eligibility for the main analysis.

LOA -2.8 to 2.94). The sign of the correlation coefficients differed between segments for 20 subjects (9.3%). The Bland-Altman plot is shown in Fig 6.

**(3) Linear regression with less stringent inclusion criteria.** 935 subjects experienced one or more events with less than 5 missing/ambiguous peaks, while still fulfilling the other inclusion criteria. Regression analysis per subject for one such event per subject gave the following results:

$SBP_{AV}$ vs. $PTT-RA_{AV}$ correlation coefficient: Median -0.91 (IQR -0.97 to -0.72)

$SBP_{AV}$ vs. $HR_{AV}$ correlation coefficient: Median 0.32 (IQR -0.26 to 0.78)

$SBP_{AV}$ vs. $PTT-RA_{AV}$ regression slopes: Median -1.07 mmHg/ms (IQR -1.6 to -0.65)

**Table 2. Patient characteristics.**

| Variable | n = 511 |
|---|---|
| Age, years [Median (IQR)] | 68 (58 to 77) |
| *Gender* | |
| Male | 302 (59.1%) |
| Female | 209 (40.9%) |
| *Hospital Outcome* | |
| Died | 87 (17.0%) |
| Survived | 404 (79.1%) |
| Unknown/Missing | 20 (3.9%) |
| *Ethnicity* | |
| White | 335 (65.6%) |
| Black | 28 (5.5%) |
| Hispanic | 16 (3.1%) |
| Other | 34 (6.7%) |
| Unknown/Missing | 98 (19.2%) |
| *Primary ICD-9 Diagnosis Group* | |
| Infectious and parasitic diseases (001–139) | 45 (8.8%) |
| Neoplasms (140–239) | 34 (6.7%) |
| Diseases of the circulatory system (390–459) | 251 (49.1%) |
| Diseases of the respiratory system (460–519) | 44 (8.6%) |
| Diseases of the digestive system (520–579) | 37 (7.2%) |
| Trauma (800–959) | 23 (4.5%) |
| Other | 57 (11.2%) |
| Unknown/Missing | 20 (3.9%) |
| *First admitted to* | |
| Cardiac Surgery Recovery Unit | 196 (40.0%) |
| Medical ICU | 111 (22.7%) |
| Surgical ICU | 79 (16.1%) |
| Coronary Care Unit | 65 (13.3%) |
| Trauma/Surgical ICU | 39 (8.0%) |
| Unknown/Missing | 21 (4.1%) |

The table displays the clinical characteristics of the subjects, as given in the MIMIC III clinical database. Instances where many values had too few instances to justify a separate category have been concocted as 'Other'. Missing data is reported per variable.

## Discussion

### Main findings

On a per-subject basis, PTT-RA$_{AV}$ displayed a moderate to strong association with SBP$_{AV}$ during rapid declines in SBP. For the majority of subjects, a clear inverse correlation was observed, as predicted by the theoretical relationship between pressure and velocity. The distribution of correlation coefficients and regression slopes did not change substantially when including an additional 424 subjects in supplementary analysis with less stringent inclusion criteria. However, for a non-negligible proportion of subjects, there was no correlation, or even a positive correlation between SBP$_{AV}$ and PTT-RA$_{AV}$. There was also a considerable range in the slopes of the regression lines, as evidenced in Fig 4. Additionally, regression slopes were not always consistent within subjects across different events. These limitations must be addressed for PTT-based tracking of BP-changes to be feasible.

**Table 3. Event characteristics.**

| Variable | n = 511 | |
|---|---|---|
| | *Median* | *IQR* |
| Duration (seconds) | 287 | (148–494) |
| SBP$_{AV}$ Max (mmHg) | 123 | (121–130) |
| SBP$_{AV}$ Nadir (mmHg) | 88 | (85–89) |
| PTT-RA$_{AV}$ Average (milliseconds) | 336 | (281–367) |
| PTT-RA, SD (milliseconds) | 6 | (4–9) |
| HR$_{AV}$ Average (beats per minute) | 87 | (76–97) |
| HR$_{AV}$ SD (beats per minute) | 1 | (0–2) |

Max, min, average and SD are calculated per subject, and the distributions of these values are displayed as median with interquartile range. SD = Standard deviation.

With regard to the spread of correlation coefficients, some outliers may be ascribed to faulty peak detection. On post-hoc manual inspection it was apparent that ECG peak detection was poor for some subjects, with bipolar pacemakers being one particular challenge for the peak detection algorithm. These segments were retained for analysis. From a physiological stand-point, one possible cause of outliers may lie in the nuances between PTT and PWV. The PTT-metric as measured from the ECG R-peak actually includes two components, namely the pre-ejection period (PEP) and the vascular transit time (VTT). The PEP corresponds to the isovolu-metric contraction of the left ventricle, while the VTT denotes the time elapsed from the pulse wave transverses the aortic valve until it appears at some more peripheral site. Principally then,

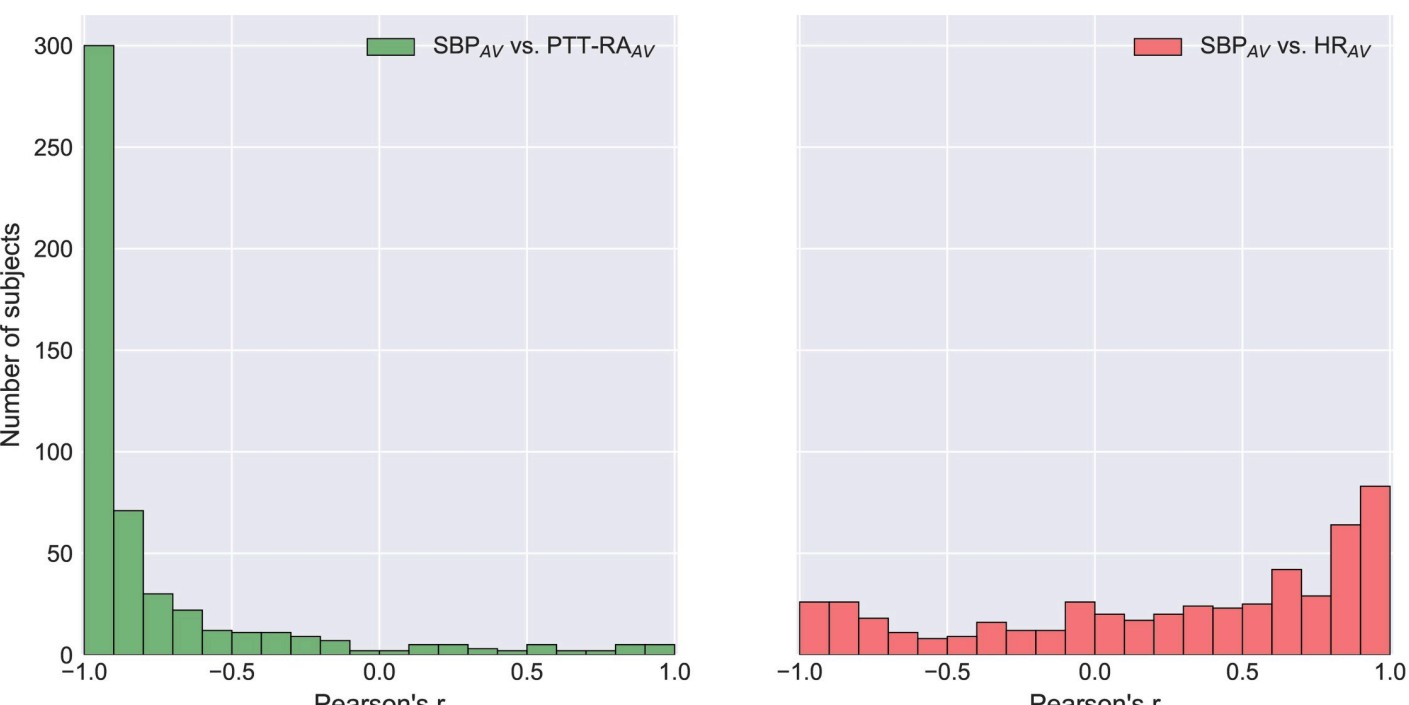

**Fig 3. Frequency histogram of Pearson Correlation Coefficients versus SBP$_{AV}$.** The figure displays the distribution of Pearson Correlation Coefficients for (a) SBP$_{AV}$ vs. PTT-RA$_{AV}$ and (b) SBP$_{AV}$ vs. HR$_{AV}$. A value of -1 signifies a perfect negative correlation, a value of 1 signifies a perfect positive correlation, and a value of 0 means no correlation. n = 511.

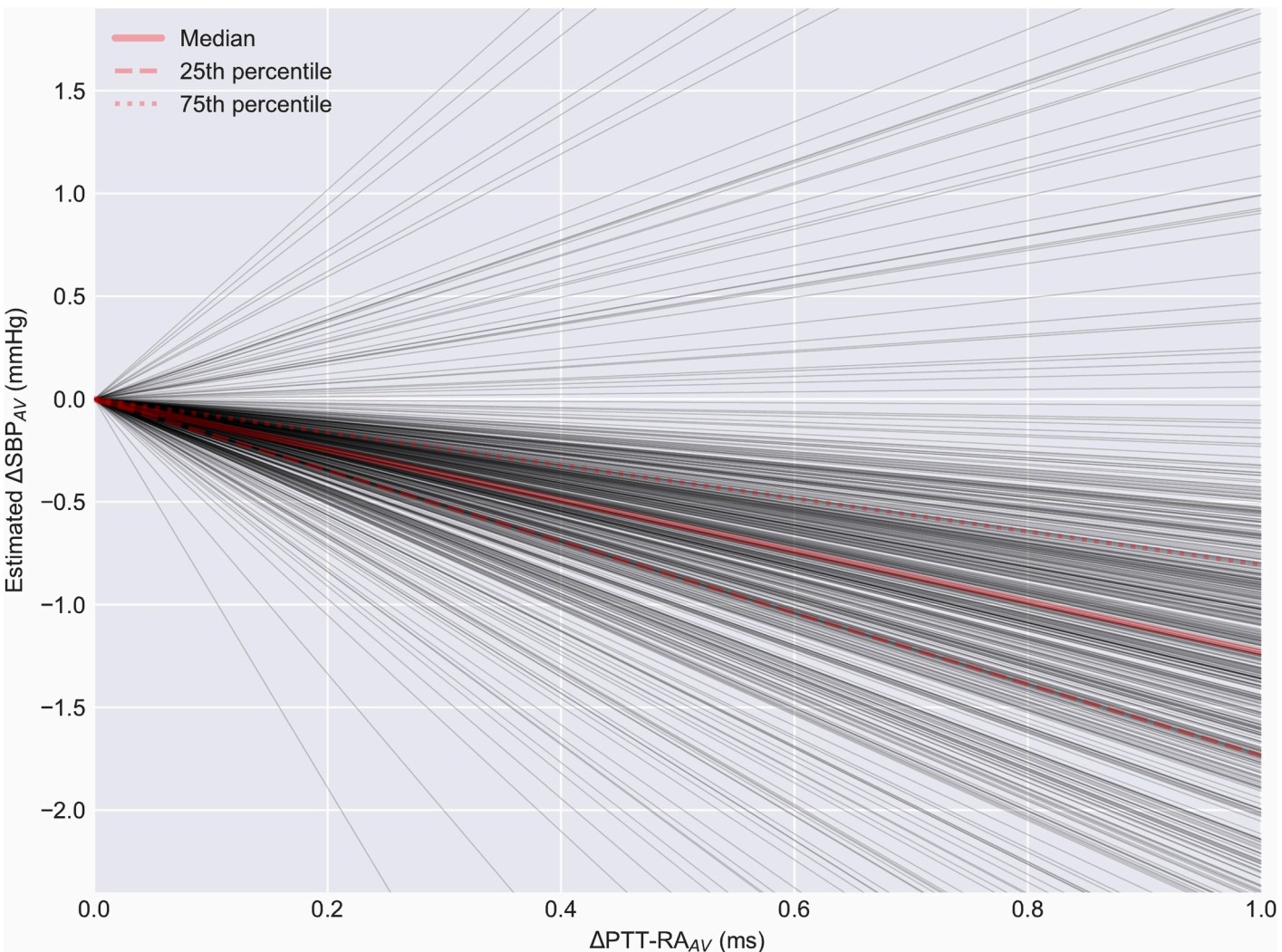

**Fig 4. Slopes from linear regression of SBP$_{AV}$ vs. PTT-RA$_{AV}$.** The figure displays the estimated change in SBP as a function of the observed PTT-RA$_{AV}$. Each dark line represents the slope of said relationship for one subject. n = 511.

VTT is the component of primary interest, as it is theoretically proportional to pressure through changes in the elastic modulus. The relationship between PEP and pressure appears to be more complex. In a series of experiments on denervated dog hearts, Wallace et al. showed that increasing HR or SV led to a shortening of PEP. Conversely, increasing aortic pressure by adjusting a mechanical resistance distal to the aorta led to lengthening of the PEP [24]. The findings of Wallace et al. suggest that changes in PEP and VTT should be equidirectional when changes in BP are mediated by changes in cardiac output, but that they would be opposite in situations where BP-changes are mediated by changes in peripheral resistance. The implication is that PTT incorporating PEP would be less robust at tracking SBP in vasodilatory shock, such as in sepsis or anaphylaxis. The etiologies of progression to hypotension in the current study are not known. However, the high proportion of subjects admitted for cardiac surgery, and the tendency of HR to be positively correlated with SBP, may suggest that the dataset is biased towards BP-reductions caused by reduction in cardiac output. Future research may focus specifically on the relationship between PTT and SBP in patients with confirmed vasodilatory shock.

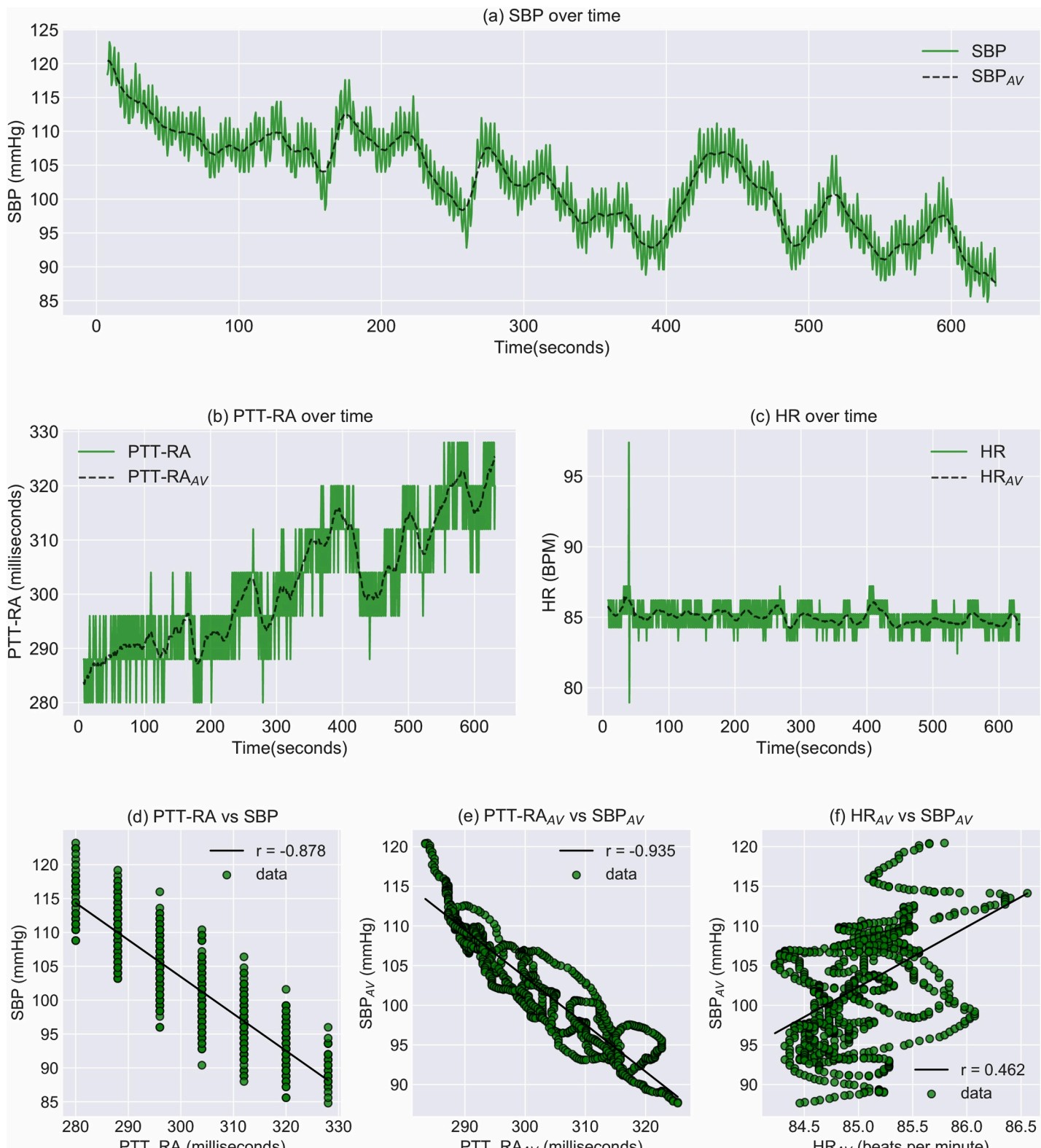

**Fig 5. Event visualization and regression for subject 20936.** a-c: Time series of extracted parameters (SBP, PTT-RA and HR). d: Regression plot of PTT-RA vs. SBP. e: Regression plot of PTT-RA$_{AV}$ vs. SBP. f: Regression plot of HR vs. SBP.

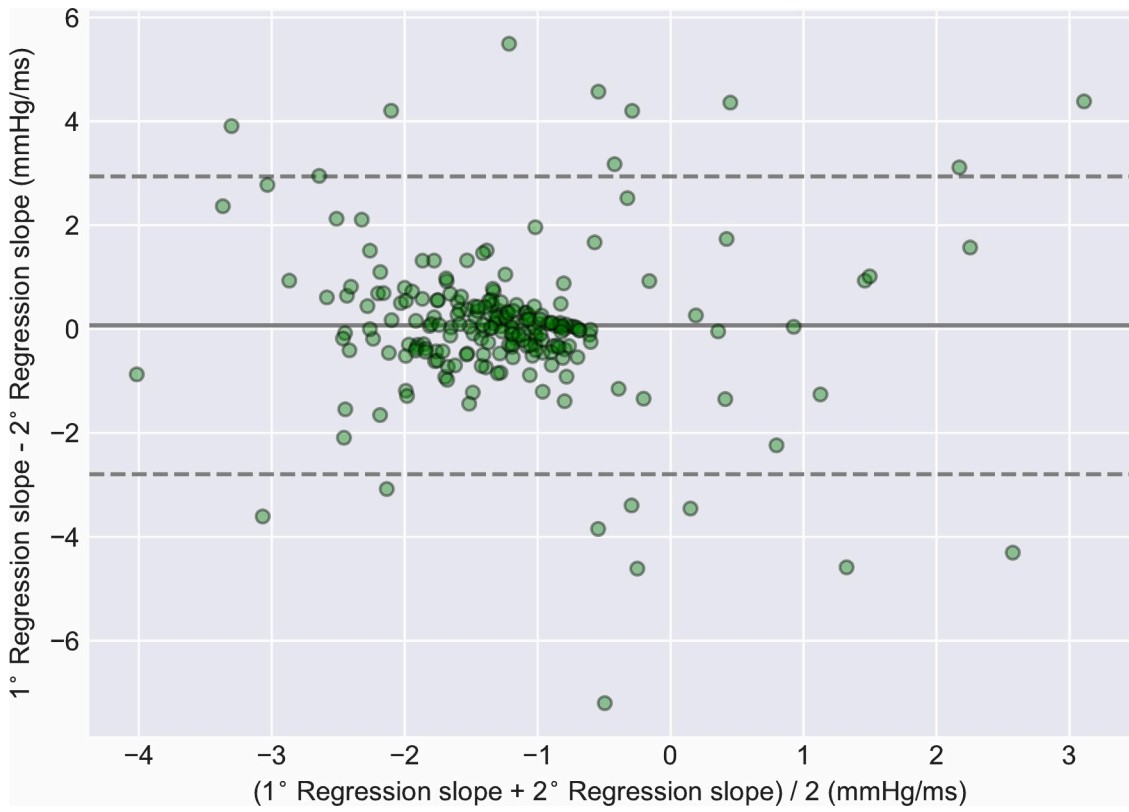

**Fig 6. Bland-Altman plot of regression slopes for primary vs. secondary segments.** The mean value of the regression slope for two segments from the same subject is displayed on the x-axis. The bias, i.e. the difference between the regression slopes of the two segments, is displayed on the y-axis (n = 215).

Averaging time series data prior to performing regression analysis may be a dubious exercise, and presents a risk of detecting spurious relationships. Generally, averaged data will always be farther from the 'truth' than the underlying data. More specifically, high frequency fluctuations in the underlying data will be attenuated. With this limitation in mind, we still believe that the use of averaging prior to regression is justified in this case, for the following reasons: I) the purpose of the current work is to model short-term trends in SBP. Beat-to-beat fluctuations in SBP, which are attenuated by averaging, are of limited interest for this purpose. II) We have included $HR_{AV}$ (averaged in an identical way) as a negative control, specifically to address the question of the method's validity. If correlations between $SBP_{AV}$ and PTT-RA$_{AV}$ were largely a consequence of averaging, similar results should be observed for the correlation between $SBP_{AV}$ and $HR_{AV}$. III) Supplementary analyses on raw data without averaging show that while magnitudes are attenuated, median correlation coefficients as well as median regression slopes display similar trends.

Having made justifications for averaging data, one should also underline the potential advantages of such an approach. Firstly, the procedure allows for the PTT-metric to assume more unique values than without averaging. When the magnitude of PTT-variability is low in relation to the sample rate, this inherently limits the strength of correlation, as demonstrated in Fig 5. Secondly, averaging reduces the influence of respiration. It is well established that respiration induces oscillations in both HR and BP. Furthermore, these oscillations have been shown to occur with a phase offset [25]. The complex interaction between respiration, HR and BP may potentially distort the beat-to-beat relationship between PTT and SBP, a problem that is avoided through

averaging. With regard to the consequence of spurious PTT-outliers, averaging presents both advantages and disadvantages. On one hand, the averaging process prevents such outliers from being interpreted as sudden extreme perturbations in SBP. On the other hand, averaging allows outliers to influence surrounding data points, causing sustained error over several heartbeats.

It is emphasized once again that the PTT metric was calculated using the peak of the ABP signal. Classically, PTT has been measured as the time interval from the R-peak of the ECG to the peak of the PPG waveform obtained at the finger. Theoretically, PTT should covary with BP through changes in the elastic modulus of the vessel, but this relationship assumes an elastic, homogenous vessel. The propagation of the pulse wave through a plethora of resistance vessels (i.e. small arteries and arterioles), in which the elastic properties of the wall are modulated by smooth muscle, is likely to confound the relationship between pressure and velocity. In light of the heterogeneity of distal perfusion status in critically ill patients, the PPG peak obtained at the finger may reasonably be inferior to that obtained from the radial artery waveform. In a practical scenario, the invasive ABP-signal is clearly not available, as that would defeat the purpose of trying to estimate SBP. As such, a non-invasive method of obtaining pulse peaks proximal to resistance vessels may be necessary for practical application of our method. For any such non-invasive method, wave morphology may be of importance, as wave rise time is included in the measured time interval.

## Novelties of the current investigation

To the best of our knowledge, this is the first investigation into the dynamics of PTT during rapid declines in systolic blood pressure. Reliability in detecting such episodes is of the utmost importance if PTT-based cNIBP is to be employed in hospital wards in the future. Furthermore, we report within-subject comparison of the PTT-SBP relationship across different episodes in time. Such data is critical to appraise the stability of said relationship, and is rather scarce in the published literature. Lastly, we provide clinical characteristics of the included subjects, which is often absent in the recent literature.

## Technical limitations

The MIMIC database has some known issues relating to waveform sampling. Signals have been downsampled to 125 Hz, limiting the resolution of the PTT and HR metrics. In this process, ECG-signals were sampled on a varying interval between 4–12 ms, instead of a constant 8 ms. Additionally, synchronicity between physiological waveforms cannot be guaranteed. These limitations should generally lead to underestimation rather than overestimation of the association between $SBP_{AV}$ and $PTT_{AV}$, but they warrant caution in interpretation of results.

## Generalizability and risk of bias

Based on the observed patient characteristics of our dataset, some reservations should be made about the generalizability of our findings. Perhaps most striking is the low proportion of non-white subjects, collectively amounting to only 15.3%. It is not known whether the relationship between PTT and BP differs substantially based on ethnicity. The median age in the cohort was 68 years, and 49.1% of subjects had a primary diagnosis of a disease of the circulatory system. It is therefore likely that there was a high prevalence of atherosclerosis among the subjects. Seeing as the theoretical relationship between BP and PTT depends on the elastic properties of arteries, and atherosclerosis may modify such properties, it is possible that our results would not be generalizable to younger, healthier individuals.

Segments to be included for analysis were selected using an automated process, which should reduce the risk of investigators introducing bias. The algorithms employed in this

process, however, require a number of cutoffs for decision rules which had to be determined by the investigators. While all such decision rules were finalized prior to data analysis, cutoffs set to reduce risk of faulty peak detection and noise could conceivably lead to a systematic bias in choice of signals. Additionally, signals which are objectively unfit for analysis due to noise and difficulty in peak detection may be associated with pathological states such as arrhythmia or poor distal perfusion. The association between SBP and PTT may be systematically different in these patients, constituting a risk of bias and reduced generalizability.

### Further research

Based on our findings, we present three potential avenues for further research:

(I) PTT-measurement must be performed non-invasively in order to have potential benefit. Using transmission-PPG limits sites to finger, toe or earlobe, but it may be desirable to obtain pulse waves from sites less affected by distal vasoconstriction. Investigators should therefore consider the use of reflectance-based PPG or other non-invasive sensor technologies to attempt to mitigate this issue.

(II) Substantial inter-individual variation in regression parameters prohibits widespread application of PTT-RA$_{AV}$ for detecting hypotension without some form of calibration or additional parameters. Further research should explore modifying factors that could aid patient selection or improve the generalizability of the method. Furthermore, it will be necessary to include patient populations in which invasive hemodynamic monitoring is not indicated, as these patients are likely the most suitable candidates for cNIBP.

(III) We have shown that PTT-RA$_{AV}$ will generally follow an inverse trajectory to that of SBP$_{AV}$ during rapid declines in SBP, indicating that the method could have high sensitivity for detecting progression to hypotension. However, as our research is limited to such episodes, we cannot comment on the specificity. Further research may aim to demonstrate whether prolongation of PTT-RA$_{AV}$ occurs during periods of stable or rising SBP.

## Conclusions

We have investigated changes in pulse transit time during rapid declines in blood pressure, in a large cohort of patients admitted to an intensive care unit. There was a moderate-strong intra-individual correlation between averaged systolic blood pressure, and averaged pulse transit time as measured from ECG R-peak to the peak of the arterial blood pressure waveform. Several methodological challenges remain, but our findings suggest that real-time, non-invasive detection of hypotension in hospitalized patients may be feasible.

## Supporting information

**S1 File. Event parameters.** Contains all the information necessary to identify the segments that were used for the main analysis.
(XLSX)

## Author Contributions

**Conceptualization:** Sebastian Grøvdal Schaanning, Nils Kristian Skjaervold.

**Data curation:** Sebastian Grøvdal Schaanning.

**Formal analysis:** Sebastian Grøvdal Schaanning.

**Funding acquisition:** Nils Kristian Skjaervold.

**Investigation:** Sebastian Grøvdal Schaanning, Nils Kristian Skjaervold.

**Methodology:** Sebastian Grøvdal Schaanning, Nils Kristian Skjaervold.

**Project administration:** Nils Kristian Skjaervold.

**Resources:** Nils Kristian Skjaervold.

**Software:** Sebastian Grøvdal Schaanning.

**Supervision:** Nils Kristian Skjaervold.

**Validation:** Sebastian Grøvdal Schaanning, Nils Kristian Skjaervold.

**Visualization:** Sebastian Grøvdal Schaanning.

**Writing – original draft:** Sebastian Grøvdal Schaanning, Nils Kristian Skjaervold.

**Writing – review & editing:** Sebastian Grøvdal Schaanning, Nils Kristian Skjaervold.

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
