## [Decision Letter · Decision Letter 0]

1 Jun 2020

PONE-D-20-12474

Rapid declines in systolic blood pressure are associated with an increase in pulse transit time

PLOS ONE

Dear Dr. Skjaervold,

Thank you for submitting your manuscript to PLOS ONE. After careful consideration, we feel that it has merit but does not fully meet PLOS ONE’s publication criteria as it currently stands. Therefore, we invite you to submit a revised version of the manuscript that addresses the points raised during the review process.

Please address all comments pointed out by the reviewers with special attention to the difference between PTT derived from ECG R - PPG peak and ECG R - BP peak (SBP).In addition, the authors should clearly mention the weakness point of former works (identification of the gaps) and describe the novelties of the current investigation to justify us the paper deserves to be published in PLOS ONE.

We look forward to receiving your revised manuscript.

Kind regards,

Kenta Matsumura

Academic Editor

PLOS ONE

Journal Requirements:

"NKS is chief medical officer and shareholder in Moon Labs, a medical-technology company that is prototyping a wearable biosensor; the company is currently not working with continuous blood pressure monitoring. SGS declares that he has no competing interests.".

i) Please confirm that this does not alter your adherence to all PLOS ONE policies on sharing data and materials, by including the following statement: "This does not alter our adherence to  PLOS ONE policies on sharing data and materials.” (as detailed online in our guide for authors http://journals.plos.org/plosone/s/competing-interests).  If there are restrictions on sharing of data and/or materials, please state these. Please note that we cannot proceed with consideration of your article until this information has been declared.

ii) Please include your updated Competing Interests statement in your cover letter; we will change the online submission form on your behalf.

Reviewers' comments:

Reviewer's Responses to Questions

**Comments to the Author**

1. Is the manuscript technically sound, and do the data support the conclusions?

Reviewer #1: Partly

Reviewer #2: No

2. Has the statistical analysis been performed appropriately and rigorously? 

Reviewer #1: Yes

Reviewer #2: No

3. Have the authors made all data underlying the findings in their manuscript fully available?

Reviewer #1: Yes

Reviewer #2: Yes

4. Is the manuscript presented in an intelligible fashion and written in standard English?

Reviewer #1: Yes

Reviewer #2: Yes

5. Review Comments to the Author

Reviewer #1: In this paper, the authors examined the correlation between pulse transit time(PTT) and systolic blood pressure(SBP) from the waveform database of an intensive care unit to monitor the rapid declines in blood pressure. The results of this study can be expected to lead to non-invasive, real-time detection of rapid declines in blood pressure. The focus of the paper is well phrased, including the problems with the method. However, I would still like make the following suggestions to hopefully improve the quality of the paper:

1. (L 59-60) More properly, the vascular unloading technique was originally proposed by Marey, and later by Shirer and Penaz (Matsumura et al., 2017, IEEE Tran. Biomed. Eng., 64(5), 1131-1137). So, the author needs to revise the phrase “originally proposed by Penaz in 1973” with adding references.

E. J. Marey, “Pression et vitesse du sang,” in Pbysiolotique Experimentale. vol. 2, Paris, France: G. Masson, 1876. Pp.307.

H. W. Shirer, “Blood pressure measuring methods,” Ire Trans. Biomed. Electron., vol. BME-9, pp. 116–125, Apr. 1962.

J. Penaz, “Photoelectric measurement of blood pressure, volume and flow in the finger,” in Proc. Dig. 10th Int. Conf. Med. Biological Eng., 1973, p. 104.

2. (L 78-79) A reference for conflicting papers should be added.

3.(L 153-154) What was the reason for using 21 samples in the averaging of SBP, PTT, and ABP?

4. (L 199-201) If the sample rate is 125 Hz, the time resolution should be 8 ms. However, why does Fig. 5(d) show a time resolution of 4 ms?

5. (Supplementary analysis(3)) Did the authors include the values of SBP,PTT-RA,HR obtained by the missing/ambiguous peaks in the calculation of SBPAV,PTT_RAAV,HRAV? If it is not included, the analysis is the same as the main analysis, and I think that it is not a proof of robustness.

Reviewer #2: The authors have investigated the relationship between declined systolic blood pressure (SBP) with pulse transit time (PTT) that was calculated from the ECG and arterial BP signals obtained in MIMIC database. The results indicate that the decrease of SBP is moderately associated with increase of PTT. Unlike the PTT calculated from ECG and photoplethysmogram (PTT), the PTT obtained from arterial BP would expect a higher correlation with BP, since it contains the direct information of BP rather than as an indirect estimate of BP. Therefore, I am not surprised with the results reported in this study. In addition, there are quite a lot study in the past decades or so on the relationship of the BP and PTT. I would therefore suggest the authors to clarify the significance of the study. My other concerns are:

1) PTT calculated from the peaks of arterial BP suffers from the reflective waveform which may impact the accuracy of the PTT;

2) I understand the synchronization issue between ECG and PPT, what about the similar issue in ECG and arterial BP?

3) For the moving average of PTT-RA, what is the time interval for the average?

4) In Fig. 6, there seems not too many data point for the estimate, but it was mentioned data from 511 subjects were included for the study

6. PLOS authors have the option to publish the peer review history of their article (what does this mean?). If published, this will include your full peer review and any attached files.

Reviewer #1: No

Reviewer #2: Yes: Xiaorong Ding

---

## [Author Response · Author response to Decision Letter 0]

23 Jul 2020

All the questions and comments from editor and reviewers are commented. They are to be found in the Response to Reviewers document in this submission.

---

## [Decision Letter · Decision Letter 1]

9 Sep 2020

PONE-D-20-12474R1

Rapid declines in systolic blood pressure are associated with an increase in pulse transit time

PLOS ONE

Dear Dr. Skjaervold,

Thank you for submitting your manuscript to PLOS ONE. After careful consideration, we feel that it has merit but does not fully meet PLOS ONE’s publication criteria as it currently stands. Therefore, we invite you to submit a revised version of the manuscript that addresses the points raised during the review process.

ACADEMIC EDITOR:

Please update Figure 5.

We look forward to receiving your revised manuscript.

Kind regards,

Kenta Matsumura

Academic Editor

PLOS ONE

Reviewers' comments:

Reviewer's Responses to Questions

**Comments to the Author**

1. If the authors have adequately addressed your comments raised in a previous round of review and you feel that this manuscript is now acceptable for publication, you may indicate that here to bypass the “Comments to the Author” section, enter your conflict of interest statement in the “Confidential to Editor” section, and submit your "Accept" recommendation.

Reviewer #1: (No Response)

Reviewer #2: All comments have been addressed

2. Is the manuscript technically sound, and do the data support the conclusions?

Reviewer #1: Partly

Reviewer #2: (No Response)

3. Has the statistical analysis been performed appropriately and rigorously? 

Reviewer #1: Yes

Reviewer #2: (No Response)

4. Have the authors made all data underlying the findings in their manuscript fully available?

Reviewer #1: Yes

Reviewer #2: (No Response)

5. Is the manuscript presented in an intelligible fashion and written in standard English?

Reviewer #1: Yes

Reviewer #2: (No Response)

6. Review Comments to the Author

Reviewer #1: The manuscript has been revised well. I think this manuscript will be acceptable after some corrections have been done.

1) According to your answer, the correct time resolution for PTT-RA is 8 ms, but the correction is not reflected in Figure 5(b), (d) and (e) of the revised manuscript.

Reviewer #2: (No Response)

7. PLOS authors have the option to publish the peer review history of their article (what does this mean?). If published, this will include your full peer review and any attached files.

Reviewer #1: No

Reviewer #2: No

---

## [Author Response · Author response to Decision Letter 1]

17 Sep 2020

We have noticed that not only figure 5 but also figure 4 and 6 and supplementary materials needed an update.

---

## [Editor Report · Decision Letter 2]

21 Sep 2020

Rapid declines in systolic blood pressure are associated with an increase in pulse transit time

PONE-D-20-12474R2

Dear Dr. Skjaervold,

We’re pleased to inform you that your manuscript has been judged scientifically suitable for publication and will be formally accepted for publication once it meets all outstanding technical requirements.

Kind regards,

Kenta Matsumura

Academic Editor

PLOS ONE
---

## [Editor Report · Acceptance letter]

29 Sep 2020

PONE-D-20-12474R2 

Rapid declines in systolic blood pressure are associated with an increase in pulse transit time 

Dear Dr. Skjaervold:

I'm pleased to inform you that your manuscript has been deemed suitable for publication in PLOS ONE. Congratulations! Your manuscript is now with our production department. 

Kind regards, 

on behalf of

Dr. Kenta Matsumura 

Academic Editor

PLOS ONE